# Research on the degree of coupling of the Internet development level and agricultural ——Ecological efficiency based on 2009–2018 data from 13 major grain-producing areas in China

Liu Shuang, Chen Ximing📀, Shang Jie*

School of Economics and Management, Northeast Forestry University, Harbin, Heilongjiang Province, China

* shangjie2005@126.com

**Data Availability Statement:** All relevant data are within the manuscript and its Supporting Information files.

## Abstract

Agricultural informatization and agricultural green development are important components of modern agricultural development, and coordination between the two is an important foundation for achieving sustainable agricultural development. This paper uses data from the 13 major grain producing areas in China from 2009 to 2018 to analyze the coordination of the Internet development level and the agricultural ecological-efficiency, and it further investigates the degree of coupling and coordination between the two. The results of theis study are as follows.(1) The Internet development level of China's 13 main grain production areas has been continuously improving. The average Internet penetration rate increased from 0.25 in 2008 to 0.54 in 2018. (2) The agricultural ecological efficiency of China's main grain production areas has gradually improved. The average value of agricultural ecological efficiency increased from 0.45 in 2009 to 0.79 in 2018. (3) The Internet development level in China's main grain production areas and the continuous improvement of coordination and degree of coupling of the agricultural ecological efficiency show that the interaction between them has led to continuous improvements in the agricultural informatization development and agricultural green development. However, the coordination between the two still has significant room for growth, and there is a certain gap between the different regions.

## 1. Introduction

Agriculture is the foundation of social development, and it directly affects human survival [1]. The development of agricultural informatization and the improvement of the agricultural ecological environment through agricultural green development are important components of the development of modern agriculture. Promoting the development of agricultural informatization and green development is an important foundation for achieving sustainable agricultural development. The application of Internet technology in the field of agriculture is an important aspect of agricultural informatization. The application of Internet technology can improve the agricultural production efficiency and increase the agricultural output, thereby providing

**Funding:** The authors received on specific funding for this work.

**Competing interests:** The authors has declared that no competing interests exist.

support for the resolution of the worldwide food crisis [2,3]. The improvement of agricultural ecological efficiency is an important manifestation of the improvement of the agricultural ecological environment, and it is also an important indicator of agricultural green development. Due to the excessive pursuit of agricultural output growth in traditional agricultural production processes, large amounts of chemical fertilizers and pesticides are used, resulting in serious environmental pollution problems [4]. The green development of modern agriculture, while emphasizing an increase in agricultural output, actively promotes the improvement of the agricultural ecological environment and improves the quality of agricultural development. Agricultural ecological efficiency is an important indicator used to measure this achievement. Research on the coordination of the Internet development level and agricultural ecological efficiency can reflect the coordination between agricultural informatization development and agricultural green development well, and thus, it provides a reference for government policy formulation and the promotion of sustainable agricultural development.

Based on previous studies, Internet development and agricultural ecological efficiency are important research hotspots that have attracted the attention of many scholars. The development of the Internet has been regarded as an important embodiment of agricultural informatization. Its important impact on agriculture and rural areas has been investigated by many scholars, and an endless stream of relevant research has emerged [5–8]. As an important indicator of agricultural green development, agricultural ecological efficiency is also an important research focus. Ecological efficiency was first proposed by the German scholars Schaltegger and Sturm [9]. The improvement of the ecological efficiency depends on the environmental impact of the lower resource consumption in the production of products [10]. Research on agricultural ecological efficiency can be a valuable reference for the formulation and improvement of government agricultural policies [10]. To this end, many scholars have discussed agricultural ecological efficiency [11–13].

As an important aspect of agricultural sustainable development, agricultural informatization and agricultural green development have important links between them. The development of the Internet has had an important positive effect on the improvement of the agricultural ecological environment. First, the application of Internet technology is of great significance for knowledge acquisition by farmers, which enables farmers to obtain important knowledge about green agriculture. Aldosari et al.'s surveys on Pakistani farmers revealed that most Pakistani farmers believe that the Internet is an effective source of agricultural information [14]. Marshall et al.'s studies in Australia also demonstrated the important role of the Internet in farmers obtaining information [15]. Second, the Internet can promote the application of advanced technologies by farmers, including water-saving technologies that are conducive to the improvement of the agricultural ecological environment. Gao et al. analyzed the impact of new Internet technology on the adoption of water-saving technology by farmers after investigating farmers in Henan Province and Shandong Province in China [16]. Their research results showed that the promotion of Internet agricultural technology improved farmers' technological adoption level to a certain extent. In addition, the development of the Internet is conducive to enhancing people's awareness of the ecological environment, and thus, it provides a better social environment for the improvement of the agricultural ecological environment. Liu showed that Internet use has a significant positive impact on environmental knowledge (EK) and the perception of the threats posed by environmental pollution (i.e., to pets) [17]. In addition, agricultural green development also promotes the development of the Internet to a certain extent, and thus, it promotes agricultural informatization. For example, agricultural green development requires the precision of agricultural production and operations, that is, reducing resource consumption through the implementation of precision agriculture. The development of precision agriculture requires the use of agricultural information

technology. Therefore, agricultural green development drives the use of agricultural information technology in agriculture.

Coupling coordination theory can be used to describe the degree of interaction between two or more subsystems, and it has been widely used in the study of coordination relationships between two systems [18]. Shi et al. analyzed the coupling coordination between economic development and the ecological environment in the tropical and subtropical regions of China [19]. Dong et al. measured the degree of coupling and coordination between the upstream, midstream, and downstream wind power industry chains in China [20]. Gan et al. measured the degree of coupling and coordination between urbanization-city-industry integration in Sichuan Province, China [21]. Chen et al. conducted research on the coordination and coupling of China's carbon emissions and ecological environment [22]. These studies demonstrated that coupling coordination theory plays an important role in the study of the coordinated development of two systems.

Research on the coordination of the Internet development level and the agricultural ecological efficiency can reflect the coordination between agricultural informatization development and agricultural green development well, thereby providing a reference for government policy formulation and promoting sustainable agricultural development. However, at present, there has not been any research that specifically explored the coordination between the Internet development level and the agricultural ecological efficiency. Thus, it is difficult to provide a reference for the formulation of coordinated regional agricultural informatization and agricultural green development promotion policies.

China is the most important agricultural country in the world [23–25]. Moreover, the Chinese government is actively promoting the development of agricultural informatization and is striving to improve the agricultural ecological environment. For this reason, it is of particular significance to analyze the coordination between China's Internet development level and agricultural ecological efficiency. Taking into account the actual development of China's agriculture, this paper takes China's 13 main grain production areas as the research area, measures the Internet development level and agricultural ecological efficiency of these main grain production areas, uses the coupling coordination model to analyze the coordination level of the two to study China's agricultural informatization and greening coordination situation, and proposes countermeasures.

The layout of this paper is as follows. The first section introduces the research background. The second section describes the research area, indicator settings, research methods, and data sources. The third section measures the Internet development level and agricultural ecological efficiency and analyzes the coordination between them. Finally, we summarize the conclusions of the study and propose countermeasures.

This article expands the research perspective and specifically explores the coordination relationship between the development level of the Internet and the agricultural ecological efficiency to reflect the degree of coupling coordination between agricultural informatization and agricultural green development. Moreover, the results of this study provide support for related policy formulation.

## 2. Materials and methods

### 2.1. Study area

In this study, 13 major grain producing areas in China (Hebei, Inner Mongolia, Liaoning, Jilin, Heilongjiang, Jiangsu, Anhui, Jiangxi, Shandong, Henan, Hubei, Hunan, and Sichuan) were taken as the research objects. The locations of the main grain-producing areas in China are shown in Fig 1.

The main reason for this is that the agricultural production in Beijing and other municipalities in China has been gradually decreasing, and the planting industry in Tibet is relatively

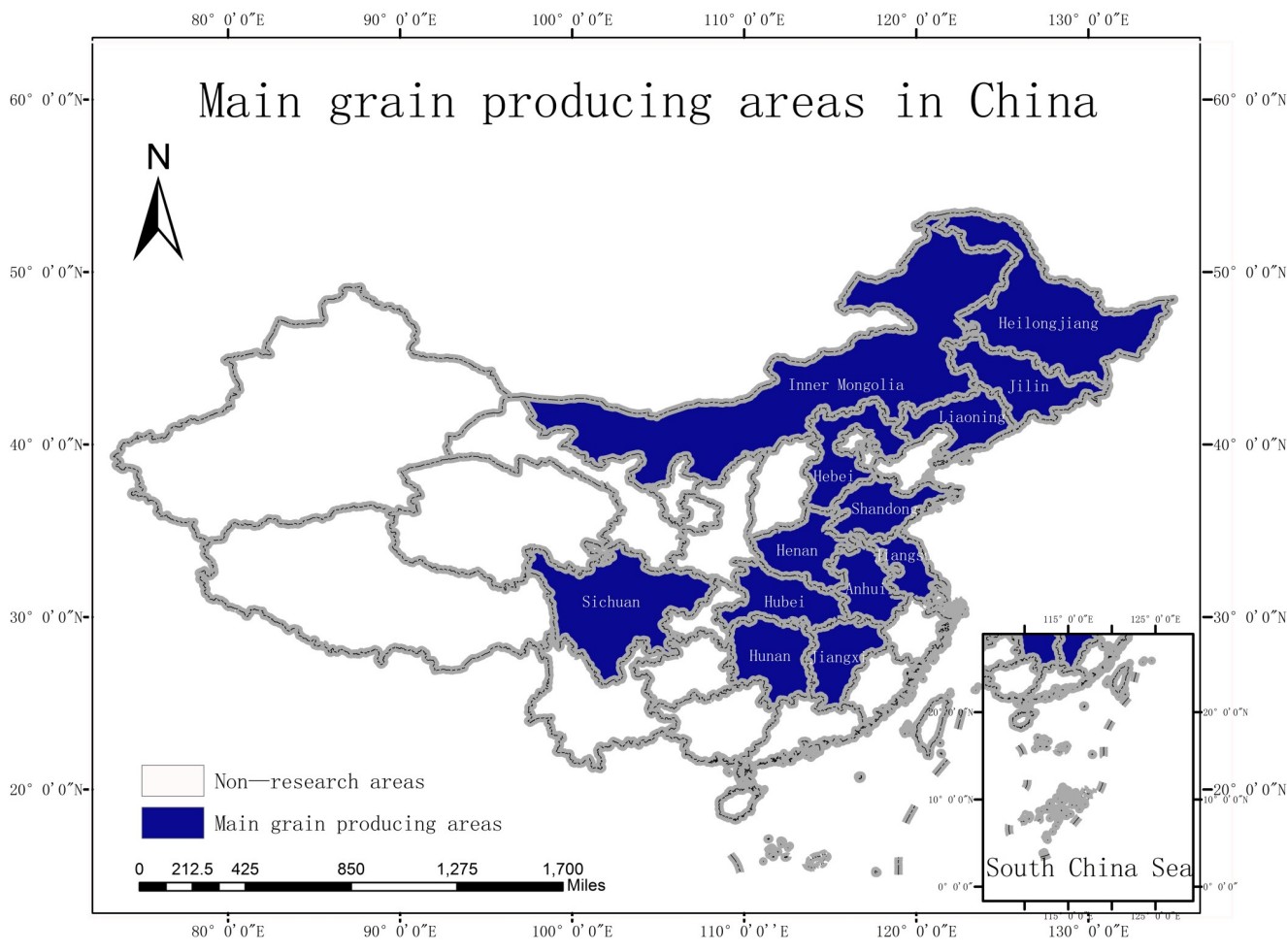

**Fig 1. Main grain producing areas in China.**

small. Therefore, by choosing the main grain producing areas with more developed planting industries as the research object, the results reflect the application of Internet technology in the agricultural field and the agricultural ecological efficiency better.

The economic and social conditions in China's 13 major grain producing areas in 2018, including the regional GDP, year-end population, gross agricultural production value, and grain output, are shown in Table 1. According to Table 1, Jiangsu Province has the highest GDP among the 13 main grain producing areas. Shandong Province is the most populous province among the 13 main grain producing areas, and it also has the highest agricultural production value. Heilongjiang Province has the highest grain output among the 13 main grain producing areas.

## 2.2. Indicators settings

**2.2.1. Internet development level.** Due to the lack of direct statistics on Internet use in rural areas and considering that the basis of Internet use is Internet popularization, in this study, we used the regional Internet penetration rate (also known as the Internet user penetration rate) to measure the development level of the regional Internet to reflect the development of the regional agricultural informatization. The regional Internet penetration rate has been used in many previous studies to reflect the level of Internet development [26,27], and the relevant equation is Internet penetration rate = number of Internet users/population.

**Table 1. Basic economic and social conditions in China's main grain producing areas.**

| | Regional GDP (100 million yuan) | Population at the end of the year (10,000 people) | Agricultural production value (100 million yuan) | Grain production (10,000 tons) |
|---|---|---|---|---|
| Hebei | 36010.27 | 7556.00 | 5707.00 | 3700.90 |
| Inner Mongolia | 17289.22 | 2534.00 | 2985.30 | 3553.30 |
| Liaoning | 25315.35 | 4359.00 | 4061.90 | 2192.40 |
| Jilin | 15074.62 | 2704.00 | 2184.30 | 3632.70 |
| Heilongjiang | 16361.62 | 3773.00 | 5624.30 | 7506.80 |
| Jiangsu | 92595.40 | 8051.00 | 7192.50 | 3660.30 |
| Anhui | 30006.82 | 6324.00 | 4672.70 | 4007.30 |
| Jiangxi | 21984.78 | 4648.00 | 3148.60 | 2190.70 |
| Shandong | 76469.67 | 10047.00 | 9397.40 | 5319.50 |
| Henan | 48055.86 | 9605.00 | 7757.90 | 6648.90 |
| Hubei | 39366.55 | 5917.00 | 6207.80 | 2839.50 |
| Hunan | 36425.78 | 6899.00 | 5361.60 | 3022.90 |
| Sichuan | 40678.13 | 8341.00 | 7195.60 | 3493.70 |

**2.2.2. Agricultural ecological efficiency measurement index system.** Based on the existing literature [28–30], combined with the actual development of China's agriculture, in this study, the agricultural labor input, agricultural land input, irrigation input, fertilizer input, pesticide input, agricultural film input, and agricultural machinery input were selected as the input indicators of the agricultural ecological efficiency. The total agricultural output value was used as the expected indicator, and the carbon emissions and pollution emissions from the agricultural production processes were used as undesirable outputs. The indicator system is shown in Table 2.

The agricultural labor force consists of the front-line personnel engaged in agricultural production. In this study, the term "agricultural practitioners" is used. Based on previous practice [31], the following formula was used to calculate the agricultural practitioners: agricultural

**Table 2. Agricultural ecological efficiency input-output index system.**

| Index type | Sub index | | Variables and descriptions | Computational method |
|---|---|---|---|---|
| Input index | Labour force | X1 | Number of employees on farm ($\times 10^4$ people) | Agricultural practitioners = first industry practitioners × total agricultural output value/total output value of agriculture, forestry, animal husbandry, and fishery. |
| | Land | X2 | Planting area of crops ($khm^2$) | Statistical Yearbook queries |
| | Chemical fertilizer | X3 | Fertilizers consumption1) ($\times 10^4$ t) | Statistical Yearbook queries |
| | Pesticide | X4 | Pesticide usage (t) | Statistical Yearbook queries |
| | Agricultural machinery power | X5 | Total power of agricultural machinery ($\times 10^4$ kW) | Statistical Yearbook queries |
| | Irrigation | X6 | Effective irrigation area ($khm^2$) | Statistical Yearbook queries |
| Output index | Total output value of farm | Y | Total output value of farm ($\times 10^8$¥) | Statistical Yearbook queries |
| Bad output index | Carbon emissions | U1 | Total carbon emissions from fertilizers, pesticides, agricultural membranes, agricultural diesel, agricultural irrigation, and agricultural sowing ($\times 10^4$ t) | Total carbon emissions = chemical fertilizer application × 0.90 + pesticide use × 4.93 + agricultural film use × 5.18 + agricultural diesel use × 0.59 + effective irrigation area × 20.48 + total crop area sown × 312.60 |
| | Pollution emissions | U2 | Quantity of chemical fertilizer, pesticide, total residues of agricultural membrane ($\times 10^4$ t) | Total pollution emissions = chemical fertilizer application × 0.65 + pesticide use × 0.50 + agricultural film use × 0.10 |

practitioners = agriculture, forestry, animal husbandry, and fishery employees (take first industry employees as a substitute indicator) × total agricultural output value/total output value of agriculture, forestry, animal husbandry, and fishery. With reference to the practices of Ning et al. [32] and Malan et al. [33], the employees in the primary industry were used as an alternative indicator of those engaged in agriculture, forestry, animal husbandry, and fishery to calculate the agricultural labor input.

Land resources are an important basis for agricultural activities. In this study, the total area of sown crops was used to reflect the land resources used in agricultural activities.

Water resources are also a necessary foundation for agricultural production activities. The effective irrigation area was used to reflect the water resources used for irrigation in agricultural activities.

Fertilizers and pesticides are important products in modern agricultural production. In this study, the input of chemical fertilizers and pesticides during agricultural activities was used to reflect the amount of fertilizer and pesticides used.

Agricultural machinery is important equipment for improving agricultural output. In this study, the total power of the agricultural machinery was used to reflect the use of agricultural machinery in agricultural activities.

The total agricultural output value reflects the results of the agricultural production activities. Therefore, in this study, the total agricultural output value was used as the expected output index of the agricultural ecological efficiency.

Various activities in agricultural production processes generate carbon dioxide. In this study, the carbon emissions from agricultural activities were considered to be an undesirable output of agricultural ecological efficiency. According to the previous research results [34–37] and the actual situation of the agricultural production in China, the carbon emissions in agricultural production activities were considered to be the total carbon emissions from the chemical fertilizers, pesticides, agricultural films, agricultural diesel, agricultural irrigation, and agricultural sowing, and the method adopted was to multiply the corresponding indicator by the emission factor. For example, the carbon emissions from chemical fertilizers = the application of chemical fertilizer × the carbon emission coefficient of chemical fertilizers. The equations used to calculate the carbon emissions from other activities are similar. The emission coefficients used in this article were obtained from the *First National Pollution Survey*: *Manual of Pesticide Loss Coefficient and Farmland Film Residue Coefficient* issued by the Chinese government. The values used are as follows: chemical fertilizers 0.90 kg/kg, pesticides 4.93 kg/kg, agricultural films 5.18 kg/kg, diesel oil 0.59 kg/kg, agricultural irrigation 20.48 $kg/hm^2$, and agricultural cultivation 312.60 $kg/hm^2$.

Improper use of chemical fertilizers, pesticides, and agricultural films in agricultural production can cause agricultural pollution. With reference to the existing research [38,39], in this study, the fertilizer pollution, pesticide pollution, and agricultural film residues were used to estimate the agricultural pollution caused by agricultural chemicals. The amount of chemical fertilizer pollution was calculated based on the data for the amount of chemical fertilizer used and the chemical fertilizer loss rate over the years—that is, amount of chemical fertilizer pollution = amount of chemical fertilizer applied × chemical fertilizer loss rate—and the equations used for the amount of pesticide pollution and the agricultural film residue rate were similar. According to the results of several domestic studies in China [40,41], a fertilizer loss rate of 65%, a pesticide pollution rate of 50%, and a film residue rate of 10% were used in this study.

*(Ethics statement: This manuscript has not been published or presented elsewhere in part or in entirety and is not under consideration by another journal. All study participants provided informed consent, and the research design was approved by the School of Economics and Management, Northeast Forestry University. We have read and understood your journal's policies, and we believe that neither the manuscript nor the study violates any of these. There are no conflicts of interest to declare).*

## 2.3. Research methods

**2.3.1. SBM model considering undesirable output.** In this study, the Slacks-based measure (SBM) model considering undesirable outputs was used to measure the agricultural ecological efficiency. The data envelopment analysis (DEA) model was first proposed by Charnes et al. [42]. When analyzing a situation with multiple inputs and multiple outputs, the DEA has unique and unparalleled advantages. The Charnes, Cooper and Rhodes (CCR) model and the Banker–Charnes–Cooper (BBC) model are the most traditional DEA models, both of which use radial and angular measurements. However, if the DUM input is too large or the output is insufficient, the use of the radial DEA model for efficiency measurement will lead to an overestimation of the efficiency of the DEA model. If there are multiple aspects of the input or output of the evaluation object, the use of the angular DEA model may cause errors in the efficiency measurement [43]. To solve this problem, Tone proposed a non-radial, non-angular, relaxation-based metric efficiency evaluation model (i.e., the SBM) [44]. The use of the SBM model to measure the efficiency of DUMk can be expressed as:

$$AEE = \min \frac{1 - \frac{1}{N}\sum_1^N \frac{S_n^X}{X_{kn}^t}}{1 + \frac{1}{M}\sum_1^M \frac{S_m^Y}{Y_{km}^t}}$$

$$s.t. \begin{cases} \sum_{K=1}^K Z_K^t X_{Kn}^t + S_n^X = X_{kn}^t, n = 1,2,3\cdots\cdots N; \\ \sum_{K=1}^K Z_K^t Y_{Km}^t - S_m^Y = Y_{km}^t, m = 1,2,3\cdots\cdots M; \\ Z_K^t \geq 0, S_n^X \geq 0, S_m^Y \geq 0, k = 1,2,3\cdots\cdots K \end{cases} \quad (1)$$

In Eq (1), $AEE$ is the efficiency; $S_n^X$ is the excessive input, $S_m^Y$ is the insufficient output, and $X_{Kn}^t, Y_{Km}^t$ are the input and output values of DUMk in period t, respectively. In practice, DUM not only has an expected output but also has an undesirable output. Tone proposed an SBM model that considers the undesirable output based on the SBM model. The SBM model that considers the undesirable output has been widely used in efficiency measurements, such as measuring energy and environmental efficiencies [45–47]. The specific form of the SBM model that considers the undesirable output is given in Eq (2).

In Eq (2), $X_{Kn}^t, Y_{Km}^t, U_{Ki}^t$ are the input value, expected output value, and undesirable output value of DUMk in period $t$, respectively; $S_n^X, S_m^Y, S_i^U$ are the redundant value of the input, the expected output, and the undesirable output, respectively. When these variables are greater than or equal to 0, they represent overuse of inputs, underproduction of the expected output, and excessive emissions, i.e., undesirable outputs.

$$AEE = \min \frac{1 - \frac{1}{N}\sum_1^N \frac{S_n^X}{X_{kn}^t}}{1 + \frac{1}{M+1}(\sum_1^M \frac{S_m^Y}{Y_{km}^t} + \sum_1^I \frac{S_i^U}{U_{ki}^t})}$$

$$s.t. \begin{cases} \sum_{K=1}^K Z_K^t X_{Kn}^t + S_n^X = X_{kn}^t, n = 1,2,3\cdots\cdots N; \\ \sum_{K=1}^K Z_K^t Y_{Km}^t - S_m^Y = Y_{km}^t, m = 1,2,3\cdots\cdots M; \\ \sum_{K=1}^K Z_K^t U_{Ki}^t - S_i^U = U_{ki}^t, i = 1,2,3\cdots\cdots I; \\ Z_K^t \geq 0, S_n^X \geq 0, S_m^Y \geq 0, S_i^U \geq 0, k = 1,2,3\cdots\cdots K \end{cases} \quad (2)$$

**2.3.2. Degree of coupling coordination model.** The degree of coupling coordination model can truly reflect the synergistic effect and the degree of coupling between the two systems, and it has been widely used. In this paper, we used this model to study the coordination between Internet development and agricultural ecological efficiency. A degree of coupling model for the Internet development level (IDL) and the agricultural ecological efficiency (ND) was constructed. The basic formula is as follows:

$$C = \left[ \frac{IDL * ND}{[(IDL + ND)/2]^2} \right]^{\frac{1}{2}} \tag{3}$$

where C is the degree of coupling between the Internet development and the agricultural ecological efficiency. Considering the coupling stability, the degree of coordination model was further introduced on the basis of the coupling model:

$$D = (C * T)^{1/2} \tag{4}$$

$$T = \alpha IDL + \beta ND \tag{5}$$

D is the degree of coupling and coordination, with a value of [0,1]. If D is close to 1, the Internet development level and the agricultural ecological-efficiency are closely coordinated. T is the comprehensive development index, and $\alpha$ and $\beta$ are the undetermined coefficients of the Internet development level and the agricultural ecological efficiency, respectively, which were set to 0.5. This weight is based on consideration of the importance of the two systems and an internal discussion of the authors' research group.

## 2.4. Data sources

The research period was from 2009 to 2018. The basic data were obtained from the *China Rural Statistical Yearbook*, the *China Statistical Yearbook*, the *Statistical Report on Internet Development in China*, and the *China Population and Employment Statistical Yearbook*.

## 2.5 Potential restricted disposal

The consistency of the data has an important impact on the research. In particular, second-hand data is prone to fluctuations. The basic data used in this article were all obtained from statistical yearbooks and related reports issued by the Chinese government. Before using these data, the statistical yearbooks issued by the National Bureau of Statistics of China were compared with the relevant data from the statistical yearbooks issued by the provincial statistical bureaus to ensure the authenticity of the data and the consistency of the national and local statistics. According to the analysis of the relevant data, the basic data is relatively stable. This is mainly reflected in the steady growth of the Internet penetration rate in each production area, the basic data for the agricultural ecological efficiency index does not exhibit significant fluctuations in any year, and the data gap between the different regions is mainly from the result of the reality of the regional natural resources and the degrees of economic and social development. Overall, the data consistency is better.

The selection of the correct model has an important impact on the research results. Based on the characteristics of the research many existing research results, the SBM model considering undesirable outputs was selected to measure the agricultural ecological efficiency and the degree of coupling coordination model was selected to calculate the degree of coordination. This model has been widely used in related research, and its advantages have been recognized by many scholars. The SBM model solves the problem that the measurement of the inefficiency

in the radial model does not include relaxation variables, but during the application process, we must pay attention to the selection of its indicators. Therefore, based on existing research results, in this study, the reality in China was used to determine the relevant indicators in order to ensure the accuracy of the efficiency measurements. During the application of the degree of coupling coordination model, the α and β weighting for the determination of the coefficients has an important impact on the results. Therefore, in this study, several experts were consulted regarding the reality to determine the best coefficients in order to improve the scientific nature of the research.

## 3. Results and discussion

### 3.1. Measurement of the internet development level

According to the statistical report on China's Internet development, from 2008 to 2018, the Internet development level of 13 major grain producing areas in China increased, and the average Internet development level rate increased from 0.25 in 2008 to 0.54 in 2018. This is an overall increase of 116%, reflecting the rapid development of China's Internet. Moreover, due to the rapid increase in the Internet development level, the application of Internet technology in agriculture has also developed a good foundation. In terms of the Internet development level in 2018, among the 13 major grain producing areas, Liaoning Province had the highest Internet penetration rate, reaching 0.63. Sichuan Province has the lowest Internet development level (only 0.48). This illustrates the development of the Internet in the different major grain production regions in China. This gap also reflects the existence of a digital divide. Figs 2–5 shows the Internet development levels in China's major grain-producing areas in 2009, 2012, 2015, and 2018. As can be seen from the figure, the Internet development level in China's major grain-producing areas has been constantly improving, especially in the areas around the Bohai Sea. Due to the relatively rapid economic development in these regions, the Internet development level has improved rapidly.

### 3.2. Measurement of agricultural ecological efficiency

This article is based on a DEA model. Based on the SBM-DEA model of unexpected outputs, combined with Coelli's research, and considering the fact that the research object of this paper is relatively macroscopic, the constant return to a scale model was selected to measure the agricultural ecological efficiency. The results are shown in Table 3. According to the results in Table 3, the results of this study are comparable to some existing research results [31,36,41], but the results are opposite to the conclusions of some experts. After comprehensive discussion, it was determined that the relevant results reflect the agricultural ecological efficiency of the main grain producing areas in China to a certain extent. According to Table 3, from 2009 to 2018, the agricultural ecological efficiency of China's 13 main grain producing areas continued to increase. The average value of the agricultural ecological efficiency increased from 0.45 in 2009 to 0.79 in 2018. It increased by 0.34, reflecting the gradual improvement of the ecological environment in China's main grain-producing areas and the continuous improvement of the agricultural output, indicating the continuous improvement of China's level of agricultural green development. However, it should be noted that although the ecological environment of China's main grain production areas has gradually improved, it has not yet reached the frontier of efficiency overall, and carbon emissions and pollution emissions still seriously affect the agricultural ecological environment.

From the perspective of the agricultural ecological efficiency in the different main grain production areas, there are large gaps between the different main grain production areas. According to the values of the agricultural ecological efficiency in 2018, the agricultural ecological efficiencies in Hebei, Liaoning, Jilin, Jiangsu, Shandong, Sichuan, and other regions have reached the frontier, but the agricultural ecological efficiency of Anhui is only 0.43.

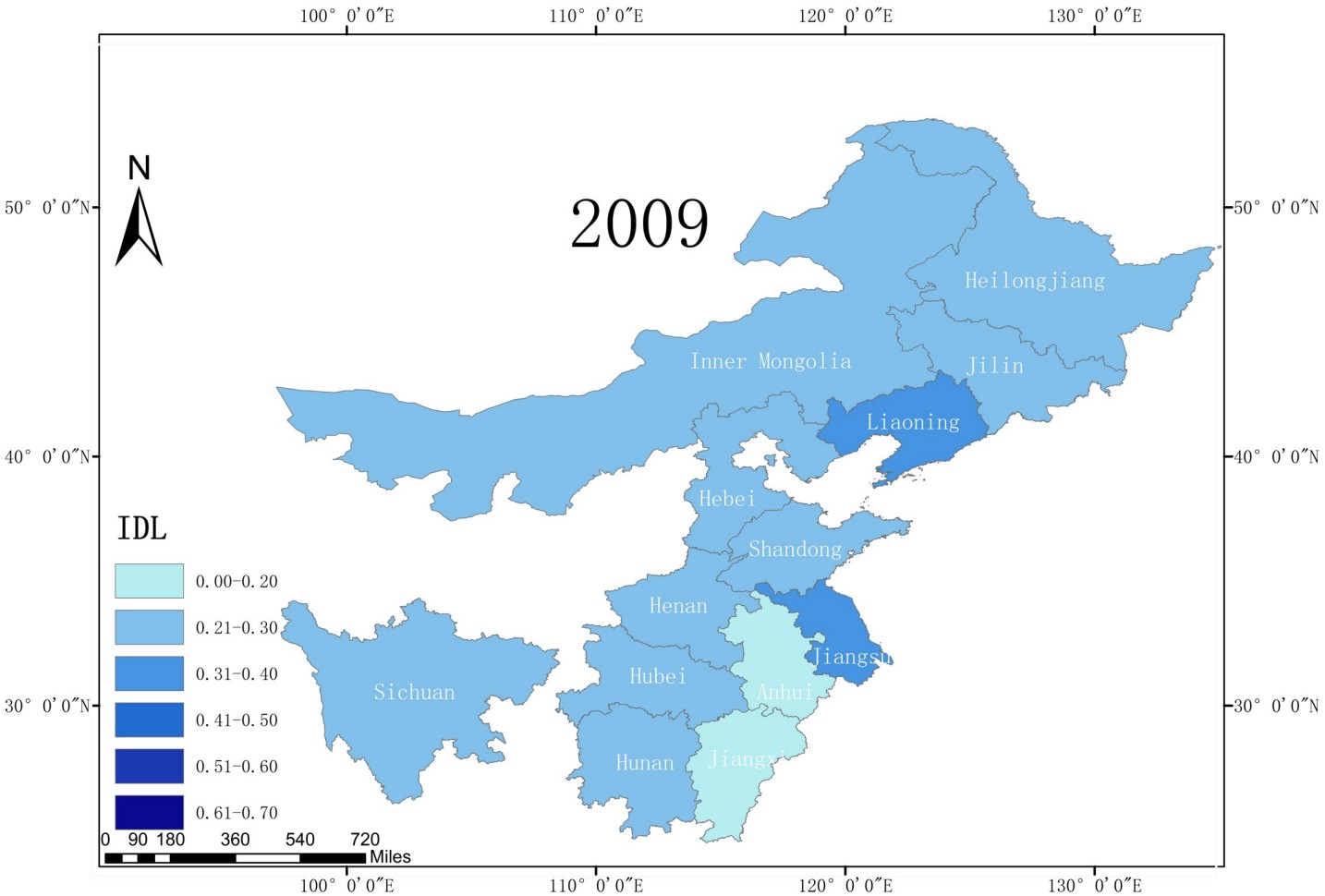

**Fig 2. Internet development level in the major grain producing areas (2009).**

Among them, Liaoning and Jilin have relatively high land fertility, so their agricultural outputs do not need to rely too much on chemical fertilizers and pesticides, and their agricultural ecological environments are relatively good. Hebei and Shandong benefit from relatively abundant rural labor, which reduces the agricultural production costs and has achieved high agricultural outputs, so its agricultural ecological efficiency is also at the forefront. Jiangsu is an economically developed area with a relatively high level of overall technology, and the agricultural output is continuously increasing due to factors such as technology. Sichuan Province is a traditional agricultural province with a dense population and good hydrological resources, which are conducive to the improvement of its agricultural ecological-efficiency. Comparatively speaking, there are gaps in land fertility, light and heat conditions, and the technical level in Anhui Province, so its agricultural ecological efficiency is low.

### 3.3. Analysis of the degree of coordination and coupling

At this stage, the application of Internet technology to agriculture to realize agricultural informatization and improve the agricultural ecological environment is an important direction for

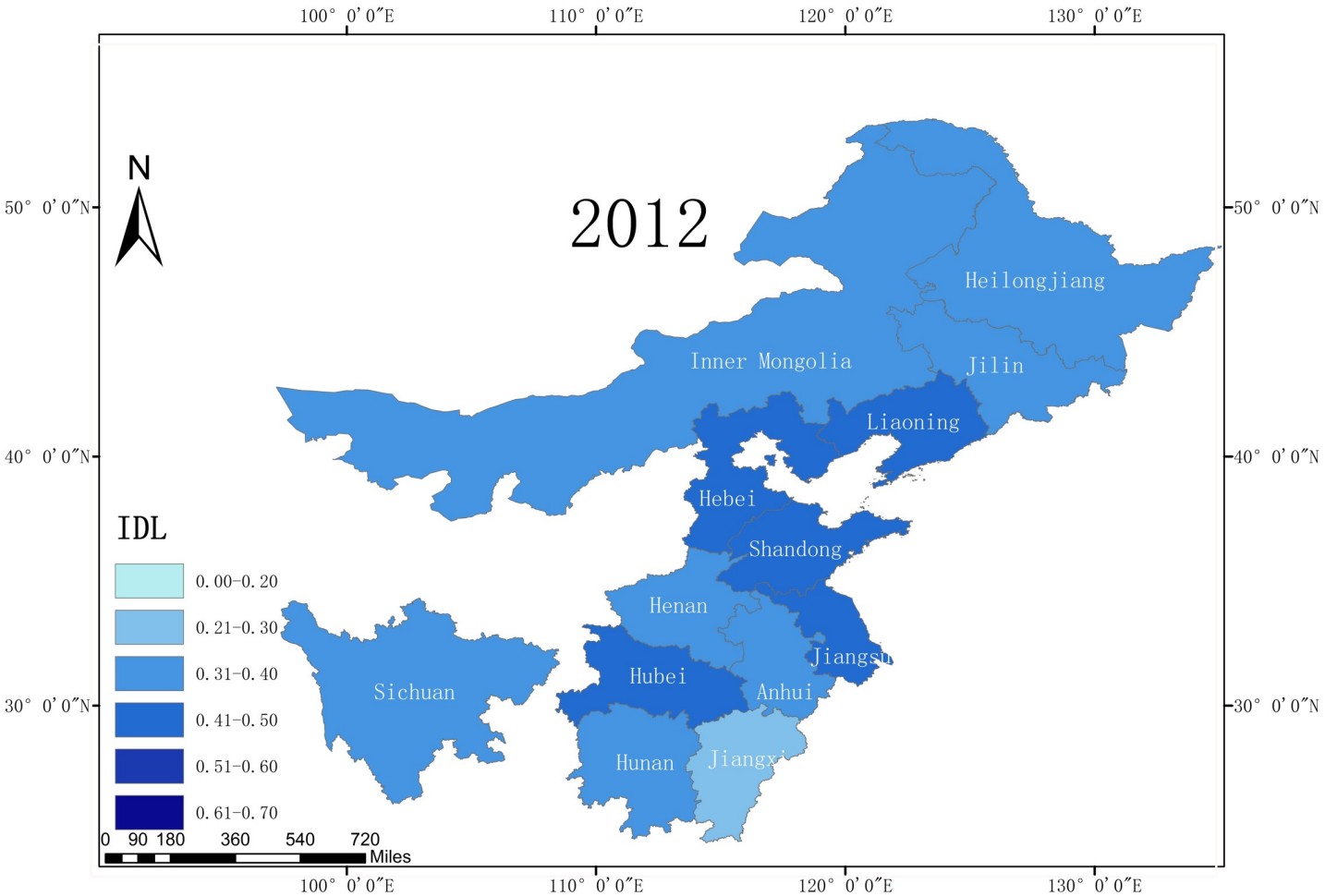

**Fig 3. Internet development level in the major grain producing areas (2012).**

agricultural development. The following is an analysis of whether the development of the two is coordinated. According to the Internet development level and the agricultural ecological efficiency obtained through the above measurements, the degree of coupling between the Internet development level and the agricultural ecological efficiency can be obtained using Eq (3) (Table 4). It can be seen from Table 4 that from 2009 to 2018, the degree of coupling between the Internet development level and the agricultural ecological efficiency in China's 13 main grain-producing areas continuously improved. The average value of the degree of coupling in all of the years was greater than 0.95, reflecting the strong interaction between the Internet development level of China's main grain-producing areas and their agricultural ecological efficiencies. However, due to the instability of the degree of coupling model during the measurement process, further calculations were carried out based on the results and using Eqs (4) and (5) to obtain the degree of coupling coordination between the Internet development level and the agricultural ecological efficiency (Table 5).

Overall, from 2009 to 2018, the mean value of the degree of coordination and coupling between the Internet development level and the agricultural ecological efficiency in China's 13 main grain-producing areas continually increased from 0.58 in 2009 to 0.80 in 2018. The

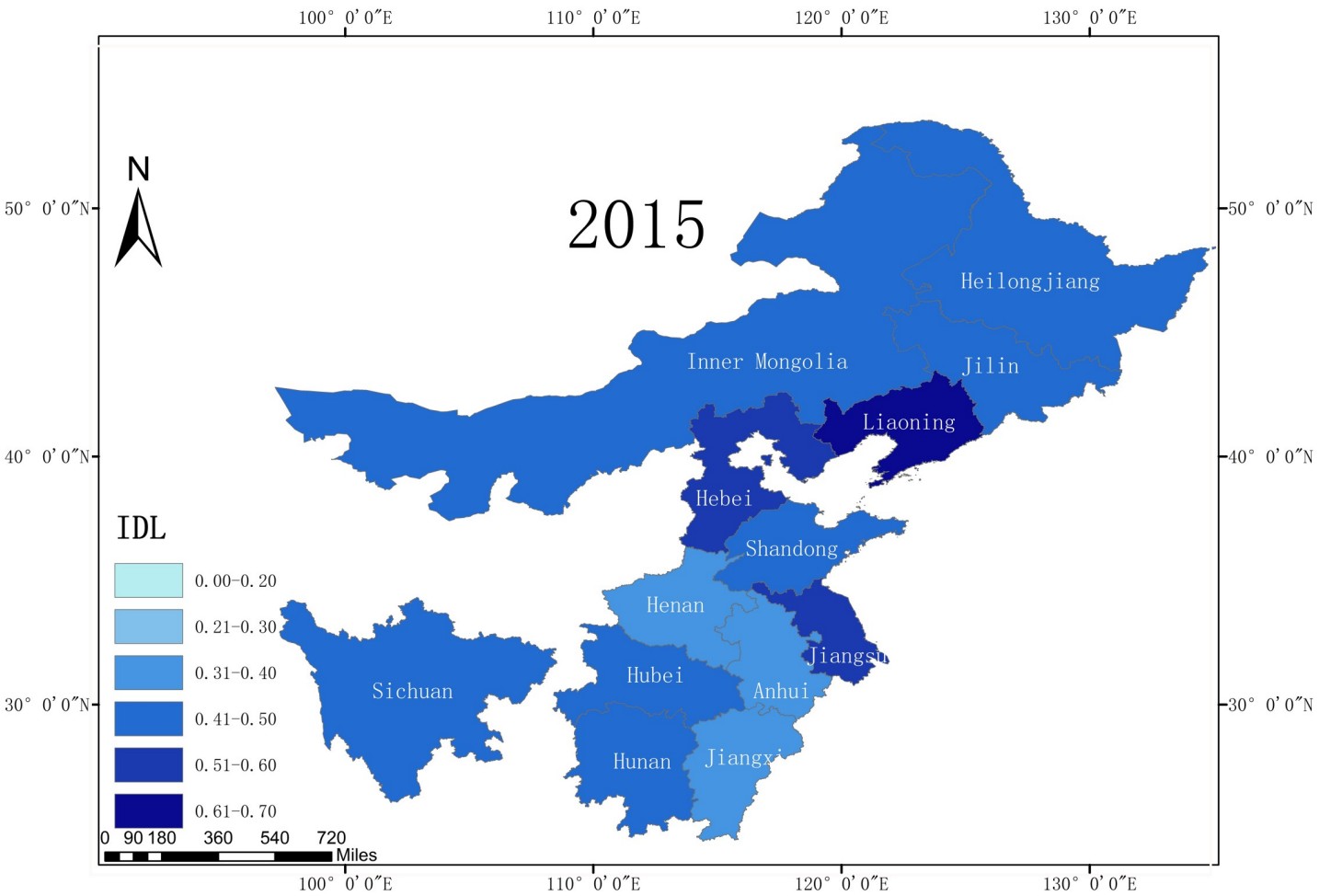

**Fig 4. Internet development level in the major grain producing areas (2015).**

improvement of the degree of coordination and coupling reflects the continuous strengthening of the influence between the two. According to the data analysis, the interactive relationship between the development of the Internet and the efficiency of the agricultural ecology in China's main grain production areas has been continuously strengthened. The improvement of the level of agricultural informatization represented by the development of the Internet is important for regional agriculture. The increases in the output and the dissemination of regional ecological and environmental protection concepts have important positive effects. With the strengthening of the interaction between the two, the positive effect of the development of the Internet on the improvement of the agricultural ecological environment will be further revealed. Based on the actual situation, in the past 10 years, the agricultural informatization in China's main grain-producing areas has continued to advance, which has greatly increased the agricultural output, and to a certain extent has improved the agricultural production efficiency and promoted the spread of environmental protection concepts. It has had a more significant impact on the improvement of the agricultural ecological efficiency. In addition, the increase in the level of agricultural green development, which is demonstrated by the improvement of the agricultural ecological efficiency, has also promoted the increase in the

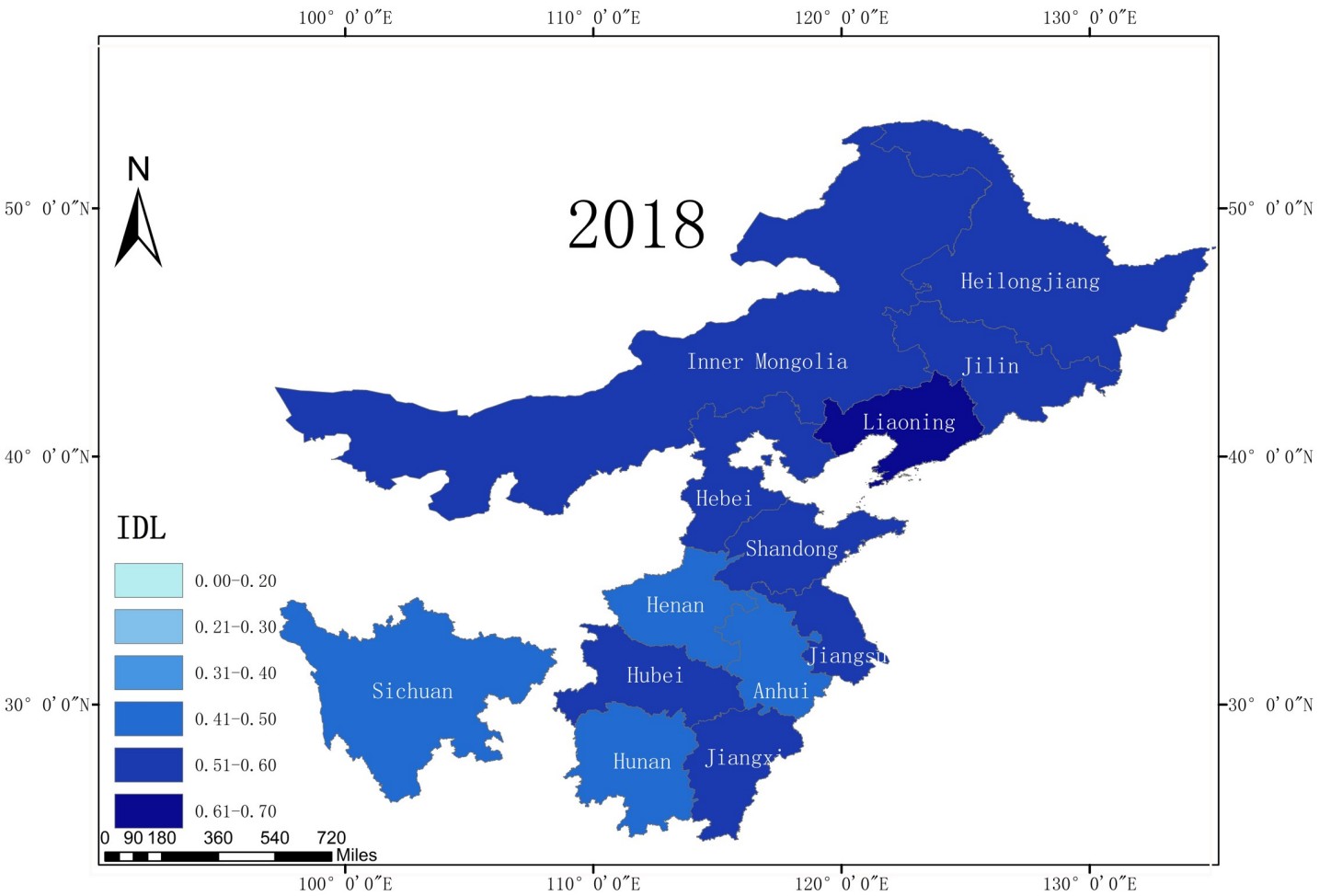

**Fig 5. Internet development level in the major grain producing areas (2018).**

income level of the agricultural operators, which enables agricultural operators to better introduce information equipment. However, it should be noted that even in 2018, the average value of the coordination coupling between the two was just over 0.80. Obviously, the coordination level between the two needs to be further improved.

From the perspective of the different regions, Liaoning Province had the highest degree of coordination and coupling between the Internet development level and agricultural ecological-efficiency in 2018 (0.89), followed by Jiangsu Province and Hebei Province. These provinces have high Internet penetration rates, and most of their agricultural ecological efficiency values are at the forefront. Internet development and the improvement of the agricultural ecological efficiency have achieved coordinated development. In 2018, the degree of coordination and coupling between the Internet development level and the agricultural ecological efficiency in Anhui Province was the lowest (only 0.68), reflecting that the interaction between the Internet development and agricultural ecological efficiency was not significant, and the development of the Internet did not effectively improve the agricultural ecological efficiency. Although the overall coordination between the Internet development level and the agricultural ecological efficiency in China's main grain producing regions is constantly improving, it

**Table 3. Agricultural ecological efficiency in the 13 major grain-producing areas in China, 2008–2018.**

|  | 2009 | 2010 | 2011 | 2012 | 2013 | 2014 | 2015 | 2016 | 2017 | 2018 |
|---|---|---|---|---|---|---|---|---|---|---|
| Hebei | 0.43 | 0.46 | 0.48 | 0.49 | 0.51 | 0.53 | 0.55 | 0.61 | 0.69 | 1.00 |
| Inner Mongolia | 0.38 | 0.41 | 0.43 | 0.42 | 0.45 | 0.45 | 0.46 | 0.48 | 0.48 | 0.56 |
| Liaoning | 0.58 | 0.58 | 0.64 | 0.67 | 0.73 | 0.73 | 1.00 | 0.84 | 0.95 | 1.00 |
| Jilin | 0.44 | 0.45 | 0.46 | 0.46 | 0.49 | 0.50 | 0.50 | 0.52 | 0.74 | 1.00 |
| Heilongjiang | 0.42 | 0.43 | 0.45 | 0.45 | 0.46 | 0.47 | 0.50 | 0.52 | 0.53 | 0.57 |
| Jiangsu | 0.57 | 0.62 | 0.65 | 0.65 | 0.69 | 0.73 | 0.77 | 0.82 | 0.90 | 1.00 |
| Anhui | 0.32 | 0.32 | 0.33 | 0.33 | 0.33 | 0.35 | 0.37 | 0.38 | 0.42 | 0.43 |
| Jiangxi | 0.33 | 0.32 | 0.33 | 0.34 | 0.44 | 0.46 | 0.48 | 0.51 | 0.55 | 0.61 |
| Shandong | 0.53 | 0.55 | 0.57 | 0.59 | 0.64 | 0.68 | 0.73 | 0.82 | 0.91 | 1.00 |
| Henan | 0.41 | 0.43 | 0.44 | 0.44 | 0.47 | 0.48 | 0.51 | 0.55 | 0.60 | 0.63 |
| Hubei | 0.44 | 0.45 | 0.46 | 0.47 | 0.49 | 0.52 | 0.56 | 0.61 | 0.67 | 0.74 |
| Hunan | 0.48 | 0.50 | 0.51 | 0.52 | 0.53 | 0.55 | 0.58 | 0.60 | 0.67 | 0.71 |
| Sichuan | 0.57 | 0.60 | 0.63 | 0.65 | 0.68 | 0.71 | 0.73 | 0.85 | 0.92 | 1.00 |
| Mean value | 0.45 | 0.47 | 0.49 | 0.50 | 0.53 | 0.55 | 0.59 | 0.62 | 0.69 | 0.79 |

should be noted that there is still room for greater growth in the coordination and coupling between the two. To this end, it is necessary to further promote the development of the Internet and to give play to the positive role of the Internet in agricultural ecological efficiency.

In terms of growth, Hebei, Jilin, and Jiangxi have seen rapid growth in the degree of coordination between the Internet development level and the agricultural ecological efficiency, increasing by 0.29, 0.27, and 0.26, respectively, in the 10 years from 2009 to 2018, showing that the Internet development and agricultural ecology in these regions have become more connected. The role of the cooperation between efficiency and the Internet has accelerated, and the coordination of agricultural informatization and agricultural green development continuously improved. The degree of coordination in Inner Mongolia and Heilongjiang has grown slowly, increasing by 0.19 and 0.18, respectively, in the past 10 years, reflecting that the cooperation between the development of the Internet and the agricultural ecological efficiency in these two regions needs to be accelerated.

**Table 4. The degree of coupling between the Internet development level and agricultural ecological efficiency in China's main grain-producing areas from 2009 to 2018.**

| Province | 2009 | 2010 | 2011 | 2012 | 2013 | 2014 | 2015 | 2016 | 2017 | 2018 |
|---|---|---|---|---|---|---|---|---|---|---|
| Hebei | 0.97 | 0.98 | 0.99 | 1.00 | 1.00 | 1.00 | 1.00 | 1.00 | 0.99 | 0.96 |
| Inner Mongolia | 0.97 | 0.99 | 0.99 | 1.00 | 1.00 | 1.00 | 1.00 | 1.00 | 1.00 | 1.00 |
| Liaoning | 0.98 | 0.99 | 0.99 | 0.99 | 0.99 | 0.99 | 0.97 | 0.99 | 0.98 | 0.97 |
| Jilin | 0.97 | 0.99 | 0.99 | 1.00 | 1.00 | 1.00 | 1.00 | 1.00 | 0.98 | 0.96 |
| Heilongjiang | 0.96 | 0.98 | 0.99 | 0.99 | 1.00 | 1.00 | 1.00 | 1.00 | 1.00 | 1.00 |
| Jiangsu | 0.97 | 0.98 | 0.99 | 0.99 | 0.99 | 0.99 | 0.99 | 0.98 | 0.97 | 0.96 |
| Anhui | 0.95 | 0.99 | 0.99 | 1.00 | 1.00 | 1.00 | 1.00 | 1.00 | 1.00 | 1.00 |
| Jiangxi | 0.96 | 0.98 | 0.99 | 1.00 | 0.99 | 0.99 | 0.99 | 1.00 | 1.00 | 1.00 |
| Shandong | 0.96 | 0.97 | 0.98 | 0.98 | 0.98 | 0.99 | 0.98 | 0.98 | 0.97 | 0.96 |
| Henan | 0.95 | 0.97 | 0.97 | 0.98 | 0.99 | 0.99 | 0.99 | 0.99 | 0.99 | 0.99 |
| Hubei | 0.97 | 0.99 | 0.99 | 1.00 | 1.00 | 1.00 | 1.00 | 1.00 | 0.99 | 0.99 |
| Hunan | 0.93 | 0.95 | 0.97 | 0.97 | 0.98 | 0.99 | 0.98 | 0.99 | 0.98 | 0.98 |
| Sichuan | 0.88 | 0.90 | 0.92 | 0.94 | 0.95 | 0.95 | 0.96 | 0.95 | 0.94 | 0.94 |
| Mean value | 0.95 | 0.97 | 0.98 | 0.99 | 0.99 | 0.99 | 0.97 | 0.99 | 0.98 | 0.98 |

**Table 5. Degree of coordination coupling between the Internet development level and agricultural ecological efficiency.**

| Province | 2009 | 2010 | 2011 | 2012 | 2013 | 2014 | 2015 | 2016 | 2017 | 2018 |
|---|---|---|---|---|---|---|---|---|---|---|
| Hebei | 0.58 | 0.61 | 0.64 | 0.67 | 0.70 | 0.71 | 0.73 | 0.75 | 0.78 | 0.87 |
| Inner Mongolia | 0.55 | 0.60 | 0.62 | 0.64 | 0.67 | 0.67 | 0.69 | 0.71 | 0.71 | 0.74 |
| Liaoning | 0.68 | 0.71 | 0.74 | 0.76 | 0.80 | 0.81 | 0.89 | 0.85 | 0.88 | 0.89 |
| Jilin | 0.59 | 0.62 | 0.63 | 0.65 | 0.67 | 0.69 | 0.70 | 0.72 | 0.79 | 0.86 |
| Heilongjiang | 0.56 | 0.60 | 0.62 | 0.63 | 0.65 | 0.67 | 0.69 | 0.71 | 0.71 | 0.74 |
| Jiangsu | 0.67 | 0.72 | 0.74 | 0.76 | 0.77 | 0.79 | 0.81 | 0.83 | 0.85 | 0.87 |
| Anhui | 0.48 | 0.52 | 0.55 | 0.57 | 0.59 | 0.60 | 0.62 | 0.64 | 0.67 | 0.68 |
| Jiangxi | 0.49 | 0.51 | 0.53 | 0.56 | 0.62 | 0.63 | 0.66 | 0.69 | 0.71 | 0.75 |
| Shandong | 0.63 | 0.66 | 0.68 | 0.70 | 0.73 | 0.76 | 0.77 | 0.81 | 0.84 | 0.87 |
| Henan | 0.54 | 0.58 | 0.59 | 0.60 | 0.64 | 0.65 | 0.67 | 0.70 | 0.72 | 0.74 |
| Hubei | 0.58 | 0.62 | 0.64 | 0.66 | 0.68 | 0.70 | 0.72 | 0.75 | 0.77 | 0.81 |
| Hunan | 0.57 | 0.61 | 0.63 | 0.64 | 0.66 | 0.68 | 0.69 | 0.72 | 0.75 | 0.77 |
| Sichuan | 0.58 | 0.62 | 0.65 | 0.68 | 0.70 | 0.72 | 0.74 | 0.78 | 0.80 | 0.83 |
| Mean value | 0.58 | 0.61 | 0.64 | 0.65 | 0.68 | 0.70 | 0.71 | 0.74 | 0.77 | 0.80 |

The fundamental reason for the above-mentioned characteristics of the Internet development level in China's main grain-producing areas and its degree of coordination with the agricultural ecological efficiency is that, to a certain extent, the degree of coupling coordination depends on the value of the degree of coupling C. According to Eq (3), the smaller the difference between the Internet development level and the agricultural ecological efficiency, the greater the value of C. When the two are equal, C reaches the maximum value of 1. Therefore, the fundamental reason for the degree of coupling in Anhui Province reaching 1 in 2012 is that the difference between the development level of the Internet and the agricultural ecological efficiency is very small, so the degree of coupling C is high. However, further analysis shows that the Internet development level and the agricultural ecological efficiency in Anhui Province are relatively low, and they only achieved a low level of coordination. In 2012, the coordination value in Anhui Province only reached 0.57. Therefore, only by simultaneously promoting the development of the Internet and the improvement of the agricultural ecological efficiency can real high-level coordination be achieved and thus can the effective coordination of agricultural informatization and agricultural green development be effectively achieved.

## 4. Conclusions and policy implications

In this study, the degrees of coordination and coupling between the Internet development level and the agricultural ecological efficiency in the 13 major grain production areas in China from 2009 to 2018 were investigated, and the main conclusions are as follows. (1) The Internet development level in the major grain production areas in China continually improved from 2009 to 2018, but an information gap still exists, and the Internet is more popular in some areas than in others. The rate needs to be further improved. (2) The agricultural ecological efficiency in China's main grain-producing areas continually improved, but the overall efficiency has not yet reached the frontier, and there is a large gap between the different regions. (3) The degree of coupling and coordination between the Internet development level and the agricultural ecological efficiency in China's main grain-producing areas continuously increased, showing that the interaction between the Internet development level and the agricultural ecological efficiency was continually strengthened, but the coordination between the two still has room for growth.

In this study, the degrees of coupling and coordination between the Internet development level and the agricultural ecological efficiency in the 13 major grain-producing areas in China from 2009 to 2018 were investigated, and the main conclusions are as follows:

1. The Internet development level of China's major grain producing areas has consistently improved, and the average Internet penetration rate increased from 0.25 in 2008 to 0.54 in 2018. However, an information gap between the major grain producing areas still exists, and the Internet penetration rate in some areas needs to be further improved. Among them, the Internet penetration rate in the major grain producing areas around the Bohai Sea is higher, while that in Sichuan Province is lower;

2. Previous studies have suggested that China's agricultural ecological efficiency is improving, but the overall efficiency has not yet reached the forefront, and there is a large gap between the different regions. The results of this study are consistent with previous research results, but most of the previous studies analyzed provinces and cities in China and did not specifically discuss the agricultural ecological efficiencies of the different grain production areas. This study further demonstrates the improvement of the agricultural ecological efficiency and the existence of regional gaps in the main grain producing areas in China. The average agricultural ecological efficiency of the 13 major grain producing areas in China increased from 0.45 in 2009 to 0.79 in 2018, reflecting the improvement of the overall agricultural ecological environment in the major grain producing areas in China. The agricultural ecological efficiency of the main grain producing areas in China is affected by their economic and natural conditions. Hebei, Liaoning, Jilin, Jiangsu, Shandong, Sichuan, and other regions have reached the forefront of agricultural ecological efficiency due to better natural conditions or more advanced technical support, while Anhui Province is restricted by its natural conditions and economic and technological development, resulting in a low agricultural ecological efficiency.

3. Previous studies have not analyzed the relationship between the Internet development level and the agricultural ecological efficiency. In this study, an in-depth investigation of this relationship was conducted. The research results show that the degree of coupling and coordination between the Internet development level and the agricultural ecological efficiency in China's main grain-producing areas continually increased from 2009 to 2018, showing that the interaction between the Internet development level and the agricultural ecological efficiency was continually strengthened, but the coordination between the two still has room for growth. The degree of coupling and coordination between the Internet development level and the agricultural ecological efficiency in China's main grain-producing areas increased from 0.58 in 2009 to 0.80 in 2018, reflecting the increasing influence of the interaction between the two. From the perspective of the different regions, the level of Internet development and the improvement of the agricultural ecological efficiency in the major grain-producing areas such as Liaoning, Jiangsu, and Hebei have achieved coordinated development, but Anhui Province has shortcomings.

In order to further promote the coordinated development of regional Internet development and the improvement of the agricultural ecological efficiency, it is suggested that the cooperation between the development of the Internet and the improvement of the agricultural ecological environment should be strengthened. (1) We should promote the application of "Internet +" methods in agricultural production and the improvement of the agricultural ecological environment; increase the promotion of "Internet +" in agricultural production and operation entities; recognize the important role of the improvement of the agricultural ecological efficiency; make full use of "Internet +" methods to promote information communication; build a

production-university-research information exchange platform for agricultural production and agricultural ecological environment development; and promote technological innovation of agricultural clean production and the dissemination of environmental protection knowledge. (2) We should actively promote the in-depth integration of the Internet and agricultural production and operations and leverage the network effect of the development of the Internet on the agricultural ecological efficiency. To this end, it is necessary to actively promote the development of digital agriculture, support agricultural production with more accurate data, and reduce the waste of resources and chemicals. In addition, we must vigorously promote the development of "Internet +" methods in various industries, so that more people can participate in the Internet, which can provide strong support for agricultural production and agricultural ecological environmental protection. (3) We should strengthen the construction of communication platforms. Convenient communication channels should be established between the different main grain production areas in China to promote the sharing of experiences in the development of the Internet and the improvement of the agricultural ecological efficiency. In addition, timely exchanges based on the coordination of agricultural informatization development and agricultural green development should be achieved.

## Supporting information

**S1 File.**
(XLSX)

**S2 File.**
(XLSX)

**S3 File.**
(XLSX)

## Author Contributions

**Conceptualization:** Liu Shuang, Shang Jie.

**Data curation:** Liu Shuang, Chen Ximing.

**Formal analysis:** Liu Shuang, Chen Ximing.

**Writing – original draft:** Liu Shuang.

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
