## [Decision Letter · Decision Letter 0]

8 Mar 2021

PONE-D-21-03106

Research on the Coupling Degree of Internet Development Level and Agricultural Eco-efficiency ——Based on the 2009-2018 Data of 13 Major Grain Producing Areas in China

PLOS ONE

Dear Dr. Ximing,

Thank you for submitting your manuscript to PLOS ONE. After careful consideration, we feel that it has merit but does not fully meet PLOS ONE’s publication criteria as it currently stands. Therefore, we invite you to submit a revised version of the manuscript that addresses the points raised during the review process.

We look forward to receiving your revised manuscript.

Kind regards,

Ghaffar Ali, PhD

Academic Editor

PLOS ONE

Journal Requirements:

2. We note that Figures 1-5 in your submission contain map images which may be copyrighted.

a. You may seek permission from the original copyright holder of Figures 1-5 to publish the content specifically under the CC BY 4.0 license. 

4. Please ensure that you refer to Figures 3, 4 and 5 in your text as, if accepted, production will need this reference to link the reader to each figure.

5. Please include captions for your Supporting Information files at the end of your manuscript, and update any in-text citations to match accordingly. Please see our Supporting Information guidelines for more information: http://journals.plos.org/plosone/s/supporting-information

Reviewers' comments:

Reviewer's Responses to Questions

**Comments to the Author**

1. Is the manuscript technically sound, and do the data support the conclusions?

Reviewer #1: Yes

Reviewer #2: Yes

2. Has the statistical analysis been performed appropriately and rigorously? 

Reviewer #1: Yes

Reviewer #2: Yes

3. Have the authors made all data underlying the findings in their manuscript fully available?

Reviewer #1: Yes

Reviewer #2: Yes

4. Is the manuscript presented in an intelligible fashion and written in standard English?

Reviewer #1: Yes

Reviewer #2: Yes

5. Review Comments to the Author

Reviewer #1: A good contribution.

The following are the comments and suggestions to the author for further improvement of the manuscript.

1. Secondary data may have consistency issues such as data gaps for certain periods or regions. Did the author notice these issues while processing the historical information of different provinces and how did the author manage if there were such cases in the data frame?

2. It is said that qualitative data always supports quantitative ones in the field of social study. Does the author conduct some FGD or KII to verify the model output results by collecting feedback from the relevant stakeholders?

3. Please insert the latitude and longitude value in the map data frame

4. Please insert the province names on the map so that reader can compare the results among different provinces

5. It would be better if the author could summarize the socio-economic differences of different provinces in a tabular form.

6. Since uncertainty is interlinked in modeling, what are the potential areas of uncertainty in the model used in this study, and how did these uncertainties of model results address?

7. Potential limitations such as consistency issues of secondary data and the uncertainty of the model used are recommended to mention in the discussion or conclusion sections.

Reviewer #2: The authors have selected an important topic and presented it in a better manner. However, I have a few minor comments:

1. Paper needs to be proofread by a native language speaker.

2. Justifications of the methods selected are weak and needs to be improved in terms of explaining / indicating academic novelty in the paper.

3. Discussion needs to be improved in terms of explaining the implication of the results in terms of both policy and academic contributions. Further, discuss the results in a comparative manner in order to explain where the results stand in academic discussions.

6. PLOS authors have the option to publish the peer review history of their article (what does this mean?). If published, this will include your full peer review and any attached files.

Reviewer #1: **Yes: **Muhammad Ziaul Hoque

Reviewer #2: No

---

## [Author Response · Author response to Decision Letter 0]

26 May 2021

Hello, dear editor:

 Thanks for your reminder, I have made the following changes：

(1)I amend the title either on the online submission form or in my manuscript so that they are identical.

(2)I ensure that you include a title page within my main manuscript. 

(3)I amend my current ethics statement to include the full name of the ethics committee/institutional review board(s) that approved my specific study.

(4)I insert my ethics statement into the beginning of the Methods section of my manuscript file. 

(5)I removed my figures/ from within my manuscript file.

---

## [Decision Letter · Decision Letter 1]

21 Jun 2021

Research on the Degree of Coupling of the Internet Development Level and Agricultural ——Ecological Efficiency Based on 2009-2018 Data from 13 Major Grain-Producing Areas in China

PONE-D-21-03106R1

Dear Dr. Jie,

We’re pleased to inform you that your manuscript has been judged scientifically suitable for publication and will be formally accepted for publication once it meets all outstanding technical requirements.

Kind regards,

Ghaffar Ali, PhD

Academic Editor

PLOS ONE

Additional Editor Comments (optional):

Reviewers' comments:

Reviewer's Responses to Questions

**Comments to the Author**

1. If the authors have adequately addressed your comments raised in a previous round of review and you feel that this manuscript is now acceptable for publication, you may indicate that here to bypass the “Comments to the Author” section, enter your conflict of interest statement in the “Confidential to Editor” section, and submit your "Accept" recommendation.

Reviewer #1: All comments have been addressed

Reviewer #2: All comments have been addressed

2. Is the manuscript technically sound, and do the data support the conclusions?

Reviewer #1: Yes

Reviewer #2: Yes

3. Has the statistical analysis been performed appropriately and rigorously? 

Reviewer #1: Yes

Reviewer #2: Yes

4. Have the authors made all data underlying the findings in their manuscript fully available?

Reviewer #1: Yes

Reviewer #2: Yes

5. Is the manuscript presented in an intelligible fashion and written in standard English?

Reviewer #1: Yes

Reviewer #2: (No Response)

6. Review Comments to the Author

Reviewer #1: Authors have revised the manuscript considering the review comments. The overall quality of the manuscript improved, and can be considered for publication in its current form.

Reviewer #2: (No Response)

7. PLOS authors have the option to publish the peer review history of their article (what does this mean?). If published, this will include your full peer review and any attached files.

Reviewer #1: No

Reviewer #2: No

---

## [Editor Report · Acceptance letter]

14 Jul 2021

PONE-D-21-03106R1 

*Research on the Degree of Coupling of the Internet Development Level and Agricultural ——Ecological Efficiency Based on 2009-2018 Data from 13 Major Grain-Producing Areas in China*

Dear Dr. Jie:

I'm pleased to inform you that your manuscript has been deemed suitable for publication in PLOS ONE. Congratulations! Your manuscript is now with our production department. 

Kind regards, 

on behalf of

Prof. Ghaffar Ali 

Academic Editor

PLOS ONE